# Optimizing Rule Weights to Improve FRBS Clustering in Wireless Sensor Networks

**DOI:** 10.3390/s24175548

**Published:** 2024-08-27

**Authors:** Jose-Enrique Muñoz-Exposito, Antonio-Jesus Yuste-Delgado, Alicia Triviño-Cabrera, Juan-Carlos Cuevas-Martinez

**Affiliations:** 1Department of Telecommunication Engineering, Universidad de Jaén, 23700 Linares, Spain; jemunoz@ujaen.es (J.-E.M.-E.); jccuevas@ujaen.es (J.-C.C.-M.); 2Department of Electrical Engineering, Universidad de Málaga, 29071 Málaga, Spain; atc@uma.es

**Keywords:** wireless sensor network, fuzzy logic, clustering

## Abstract

Wireless sensor networks (WSNs) are usually composed of tens or hundreds of nodes powered by batteries that need efficient resource management to achieve the WSN’s goals. One of the techniques used to manage WSN resources is clustering, where nodes are grouped into clusters around a cluster head (CH), which must be chosen carefully. In this article, a new centralized clustering algorithm is presented based on a Type-1 fuzzy logic controller that infers the probability of each node becoming a CH. The main novelty presented is that the fuzzy logic controller employs three different knowledge bases (KBs) during the lifetime of the WSN. The first KB is used from the beginning to the instant when the first node depletes its battery, the second KB is then applied from that moment to the instant when half of the nodes are dead, and the last KB is loaded from that point until the last node runs out of power. These three KBs are obtained from the original KB designed by the authors after an optimization process. It is based on a particle swarm optimization algorithm that maximizes the lifetime of the WSN in the three periods by adjusting each rule in the KBs through the assignment of a weight value ranging from 0 to 1. This optimization process is used to obtain better results in complex systems where the number of variables or rules could make them unaffordable. The results of the presented optimized approach significantly improved upon those from other authors with similar methods. Finally, the paper presents an analysis of why some rule weights change more than others, in order to design more suitable controllers in the future.

## 1. Introduction

Nowadays, wireless sensor networks (WSNs) are experiencing increasing importance due to Internet of Things (IoT) applications [1], which gather information from a wide variety of environments, such as health monitoring [2], smart cities [3], industrial facilities [4], vehicles [5], greenhouses [6], etc. These WSNs are composed of tens or hundreds of small devices called nodes or motes [7]. These nodes have limited processing and storage capabilities due to the fact that they are usually isolated and battery powered, as well as needing a wireless interface to communicate to the base station or directly to the Internet through a gateway using, e.g., LoRaWan technology [8].

As commented previously, nodes are commonly powered with batteries because most WSN applications do not allow a power connection to the electric grid. Moreover, those batteries are usually difficult to recharge, so efficient resource management is required in every node, mainly when they have to send data, which is the most energy-demanding process in a node. Among the several techniques used in WSNs to achieve efficient resource management (e.g., duty cycle scheduling [9] and data aggregation [10]), clustering appears one of the most important due to its capability to reduce the distance to send data [11]. Thus, with clustering it is possible to considerably decrease the energy spent in communications, and it can also be applied with other techniques like data aggregation.

Clustering is based on the premise of selecting some nodes among the total of the WSN. These nodes are called cluster heads (CH) and they are selected based on an algorithm. The remaining nodes choose one of those CHs to send their data to them. After receiving information from normal nodes (e.g., temperature, humidity, gas concentration, etc.), a CH aggregates these data and sends them to a base station (BS) or to a new layer of CHs when the network has a set hierarchy [11]. This process of selecting a set of CHs is commonly known as a round. A round consists of selecting the CHs, gathering information from the nodes via the CHs, and the transmission of the data from the CHs to the BS. In centralized clustering methods, it is usual that the BS initiates each round, whereas in distributed methods this process usually requires some kind of scheduling, due to the lack of a common reference.

Considering how clustering can be applied to the management of WSNs, it can be developed in several ways. Consequently, Figure 1 shows some of the most common approaches for clustering of WSNs.

According to Figure 1, the first level in the classification takes into account how long the current selection of CHs is maintained. Thus, the skip parameter defines how many rounds will have the same CHs, avoiding the selection of these nodes in the second and ongoing rounds. Therefore, when the skip parameter is greater than one, the whole network can save the energy of selecting CHs, which always requires sending new control messages. Thus, as far as skip is concerned, most reviewed works renew the nodes that are selected as CHs each round; therefore, this is equivalent to employing a value of one for skip, so that new CHs are selected for each round [12].

Sometimes, the skip value changes over time depending on the network conditions. In [13], the authors defined a skip based on the output of a fuzzy controller. In [14], the authors proposed a semi-distributed method where the BS defines the skip value, while the CHs are selected by a fuzzy controller whose inputs are statistically normalized based on the data gathered from each node. A variable skip value was also computed in [15]. In this approach, nodes become CHs depending on prediction of the energy waste of being a CH.

The next level in the taxonomy in Figure 1 refers to the exact point at which the decision of selecting a node as a CH is performed. Thus, if the algorithm that determines the CH selection runs at a unique point like the BS, this is a centralized method. Conversely, if each node promotes itself as a CH depending on an internal processing that considers the data gathered by that node, the clustering is defined as a distributed method. Thus, in cases where the awareness of the whole network is high, centralized clustering algorithms usually perform significantly better than distributed ones. However, the acquisition of data often requires more complex processes of information interchange and the use of additional hardware like GPS devices in the nodes to obtain their precise location.

The way in which centralized and distributed clustering methods select CHs can be achieved by stochastic or heuristic methods or by some kind of artificial intelligence technique like fuzzy control or genetic algorithms, among others. For example, ref. [16] presented a centralized method that uses two heuristic equations to classify nodes: the first is based on coverage and the second on node connectivity. In [17], the authors used a particle swarm optimization (PSO) strategy to select the CHs. In addition, machine learning techniques are widely applied to WSN management, as pointed out in [18]. Fuzzy rule-based systems (FRBS) are the most common approach in these kinds of applications. In [19], the CH candidates were sorted based on a Type-II fuzzy system. In [20], an expert system was used with three variables to choose the best nodes to become CHs. Finally, as can be observed in Figure 1, the BS in a WSN application can be in a fixed location or it can move along a sensing path to cover all the nodes in the network. It must be noted that BS mobility allows more efficient data gathering, because it can be closer to the contributing nodes or CHs. With this approach, they will spend less energy and it also can cover a larger area [21]. However, the visits of the BS to a group of nodes must be carefully synchronized with the instant that they are ready to send data and considering the distance that the BS has to move.

As was previously mentioned, fuzzy control systems are usually applied in clustering methods [22]. However, the design of the knowledge base is a demanding task because it depends on the expert knowledge of the research team. In addition, when the number of variables and fuzzy sets are high, the definition of precise rules becomes a challenging problem. This often results in rules that have minimal effect in the developed method, while they waste computational time in the fuzzy engine. Moreover, in the case of applying clustering for WSN lifetime maximization, there are three instants that are usually studied [23]: the instant when the first node depletes its battery (or dies), called first node dies (FND); the round when half of the nodes are dead, or half of the nodes have died (HND); and the moment when the last alive node dies, or last node dies (LND). Thus, when a clustering method tries to maximize those parameters, one may be improved but the other two parameters could be negatively affected. This is a common problem of multi-objective optimization in WSNs [24] (the Pareto front), which degrades WSN performance. Consequently, a research team should make certain trade-offs according to the application they are studying. In particular, they should choose which of the parameters (FND, HND, or LND) is more important for the application. Therefore, in this paper, we present the optimization of the knowledge base of a Type-1 fuzzy control system adapted to the particular performance metric (FND, HND, or LND) that is expected to be improved. The proposed clustering method was applied in a WSN that was deployed in a rectangular area whose skip value was equal to 1. The clustering method employed was centralized with a BS in a fixed location for the entire duration of the application.

The main contribution of the present research paper is the definition of a centralized fuzzy controlled clustering algorithm ruled by three stages, as a result of a PSO optimization process or CFC3PSO. In each of the aforementioned stages, a different knowledge base is used, which, as mentioned above, is the product of an optimization process. According to our previous research, the application of three different KBs, which are the result of an incremental optimization performed by a PSO system, represents a certain innovation in the field of clustering. Therefore, the reason for using the presented optimization process to improve the performance of a WSN in its three phases of operation (up to FND, from FND to HND, and from HND to LND), taking into account three possible deployments of the base station, was because we wanted to achieve a clustering algorithm that can adapt to different deployments of the WSN, without having to make adjustments. Thus, the knowledge bases used in the fuzzy system for the selection of the CHs are prepared for situations in which the BS is inside the deployment area, as well as if the BS is in its vicinity, or even far from it. Consequently, this clustering method was developed using a optimization process based on a PSO that detects which rules of the entire knowledge base are meaningful for the clustering method in order to maximize the value of one of the three parameters (FND, HND, and LND). Moreover, as a consequence of the optimization process, the processing in the BS can be faster, because less relevant rules can be omitted from the fuzzy control system. Thus, after the optimization process, each rule is assigned a weight for one of the performance metrics (FND, HND, and LND). If the WSN application needs, for example, a highest LND, only the set of fuzzy rules with high weights for that parameter are chosen, while the rules with low weight values will barely affect that application and, consequently, they will be removed. In addition, the optimization process can detect those rules that are not meaningful for maximizing any of the three parameters, which helps with the analysis of the design process and can result in modifications that improve the system, such as removing, adding, or tuning fuzzy sets; suppressing or adding fuzzy variables; etc.

This paper is organized as follows: Section 2 presents the optimization algorithm for the rule base in a clustering application for WSNs. Then, in Section 3, the results of the proposed algorithm are shown within a specific scenario, while, in Section 4, we discuss how to interpret the final rule base and the weights obtained for the rules. Finally, the paper ends with Section 5, where the conclusions of the present work are presented.

## 2. Materials and Methods

This section describes the optimization process that determines the weights to be assigned to each rule in the knowledge base of a fuzzy control system used for a clustering method in a WSN. The optimization process is performed by a PSO algorithm, which will be described later in this section. The section is structured as follows:A brief description of fuzzy logic: some concepts related to fuzzy logic will be introduced, for example, input and output variables and knowledge base (see Section 2.1).The clustering algorithm developed for the present approach is detailed (see Section 2.2.)The energy model for radio communications between nodes in the WSN is described (see Section 2.3).The simulation scenarios used for the optimization process (area dimensions, number of nodes, mode of deployment, etc.). This subsection also includes a brief description of the network model used in the simulations (see Section 2.4)The optimization process based on PSO is described (see Section 2.5).

It should be noted that the optimization process simulates different WSN deployments in three scenarios to obtain the final weight of the rules. Consequently, these simulations need an energy model (see Section 2.3) to evaluate the energy consumed by the nodes in their communication and internal processing (e.g., data aggregation).

### 2.1. Description of the Fuzzy System

Fuzzy logic [25,26] is applied to numerous scientific fields and to engineering in multiple types of applications, including denial-of-service cyber-attacks [27], flying ad hoc networks [28], microgrid systems [29], electric vehicles [30], and earthquake control [31], among others. The well-known structure of a fuzzy controller can be observed in Figure 2. A fuzzy controller is characterized by a knowledge base (KB) that contains the fuzzy rules that relate the input variables to the output variables. These rules are formed by a series of linguistic variables whose values represent the design of the fuzzy sets for any quantity or information that the fuzzy controller needs. In order to use the value of an input variable in the fuzzy controller, its crisp value must be first fuzzified, transforming this numerical value into a membership degree of the different fuzzy sets. These fuzzy values are then put together using the rules of the KB to obtain the fuzzy values of the output variables, which are defuzzified to obtain the final output of the fuzzy system. For this proposal, we decided to use a Mamdani system, so that the final output is the centroid of the aggregated output fuzzy sets.

Taking into account the fuzzy system designed for this paper, we selected some of the most common input variables used by other authors for clustering in WSNs. It must be also stated that the design of the fuzzy sets for all of these variables was the same, defining three triangular shapes (because they were for a Type-1 fuzzy system), as depicted in Figure 3. Thus, the input variables used in the present optimization process were the following:Distance to the base station (dBS). This input is considered in most fuzzy controllers [32,33] because it informs about the dispersion of the nodes. The degree of dispersion is critical when nodes are only powered by batteries. This variable measures the normalized distance of each node to the BS using the maximum distance of a node to the BS as the normalization parameter.Centrality (*C*). This variable is calculated based on the distance of the nodes in a CH to the center of the sensing area of their cluster [20]. In this approach, we use the Kmeans [34] function to find the center of the every area, and then the distance from each node to the center of that area is calculated.Node degree (ND). This parameter is the number of neighbors that each node has. We assume that a neighbor is a node that is closer than $d_{0}$ meters ($d_{0}$ is a constant of the radio model that will be detailed later (see Equation (Equation 3)) [35]. The neighbors of each node are then normalized by the maximum number of neighbors reached by one node in the set of the complete WSN.Residual energy (RE). This value is used in most clustering methods for WSNs [36,37]. It is normalized by the maximum initial energy of the nodes, which is assumed to be equivalent for all of them.CH rotation (CHR). This input variable informs about the number of times a node has been selected as a CH [38]. The reason for incorporating this variable into the fuzzy system is to limit the intensive use of energy in a node when it is selected as CH in continuous or very close rounds. In this approach, CHR is calculated as in Equation (Equation 1), where timesCH is the number of times a node has been selected as the CH. Every time a node is promoted to CH, timesCH is incremented by one and then normalized by the maximum value of this parameter for each round.
(1)$$CHR=1-\frac{timesCH}{max(timesCH)}$$

Following with the design of the fuzzy system, all input variables except dBS are defined by linguistic values: low (l), medium (m), and high (h). In the case of dBS, the fuzzy sets are named close (c), adequate (a), and far (f). The exact design of the fuzzy sets for the input variables is depicted in Figure 3.

Regarding the output of the proposed fuzzy control system, it only has one variable, defined as the chance of a node becoming a CH. Therefore, if the conditions that surround a node are propitious, the fuzzy system will infer a high chance, whereas in other cases, the chance obtained will reduce the possibilities of a node becoming a CH. It must be noted that the proposed clustering method is centralized. This means that all the executions of the fuzzy engine are accomplished by the BS, so that, for each round, the BS evaluates every node, executing the inference engine with the values for the input variables and obtaining the chance for all the nodes (the selection algorithm will be detailed in Section 2.2). Nevertheless, the output variable chance has a more complex layout that the input variables, being defined by 11 triangular fuzzy sets, as can be observed in Figure 3. The names of the fuzzy sets are as follows: very very weak (vvw), very weak (vw), weak (w), little lower medium (llm), lower medium (lm), higher medium (hm), medium (m), little higher medium (lhm), little strong (ls), strong (s), very strong (vs), and very very strong (vvs).

We defined 11 fuzzy sets for the output variable, because the number of output fuzzy sets depends, to some extent, on the number of rules. Therefore, when there are many rules, the number of fuzzy sets should be increased. When the number of inputs is small and has few sets, there are few rules and the output is usually in five sets, when the number of inputs is increased, the output is usually increased. In [39], the fuzzy system used four inputs, and consequently the output variable was composed of seven fuzzy sets. In our case, by increasing the number of inputs to five, we decided to increase the number of fuzzy sets of the output to 11, thus also increasing the variability, which allows greater adaptability in the optimization process. In particular, triangular sets were used because they also provide an immediate solution to the optimization problems that arise in fuzzy modeling [40].

Continuing with the description of the fuzzy control system, the knowledge base, as was previously mentioned, binds the inputs and the output through a set of linguistic rules that were designed by the research team based on their expert knowledge. For this approach, there are five input variables defined by three fuzzy sets each. As a result, the total number of rules to define is $3^{5}=243$ rules. All these rules were stored in a Excel file and can be consulted at the published dataset (See data availability statement). As an example, an excerpt of the rules can be found in Table 1.

As can be observed in Table 1, the initial weight of the three rules is set to one (as for all the other rules not shown initially). Then, as detailed later in the paper, these weights are optimized for three different BS locations to obtain a general approximation for different types of scenarios.

Returning to the rules presented in Table 1, the first rule represents the situation of a node that is close to the BS but with a low centrality and node degree. This means that the node has poor connectivity. It also has a low energy and has been moderately selected as a CH. Consequently, this node will have a medium probability (llm). In the case of the second rule, the node is placed in an adequate location, it also has a good connectivity (*C* and ND high), but its energy is decreasing and it has been selected several times previously as a CH. Thus, the output probability will be better than the first rule, giving a value of ls. For the third rule, although the node was placed far from the BS, all other variables are sufficiently good, so the chance value should be strong (s).

### 2.2. Clustering Algorithm

After describing the fuzzy system designed for the clustering method CFC3PSO, it is necessary to detail the algorithm that governs the CH selection in the whole WSN. As mentioned before, the proposed clustering method is centralized. Consequently, the BS has to perform all the processing and then communicate the results to the nodes. In order to minimize the information exchanged between the nodes and the BS, the BS calculates or estimates all the values it needs from the nodes, because some of them are then related to the node location (dBS, *C*, ND), which needs to be communicated or estimated only once based on the received signal strength indicator (RSSI). In the same way, the BS knows the history of each node, so it can perfectly know CHR. Thus, the unique information that the BS has to calculate based on an estimation of the energy used by each node in each round is the residual energy RE. Therefore, with all these values, in each round (skip=1), the BS performs the following steps:The BS calculates or estimates the input values for every alive node in the network.The BS runs the fuzzy controller for every node to obtain its chance.The chances are sorted from the maximum value to the lowest.The BS selects the *N* nodes with the highest probability value, which become CHs. The value of *N* is determined by the optimum number of CHs defined for LEACH [12], calculated as N=0.05∗T, with *T* being the total number of nodes.The BS sends a broadcast message with the selected CHs.Contributing nodes send their data to the closest CH.Each CH aggregates the received data and sends them to the BS.

Obviously, all the tasks in the previous algorithm assume an energy cost that is mainly consumed by the BS. However, all the alive nodes have to receive and send a message per round, so the selection of the right CHs is of paramount importance to reduce the transmission distances and consequently the energy consumed in the communication. Therefore, the right selection of the CHs will increase the lifetime of the WSN and its application can be carried out for a longer time. Thus, in order to evaluate the lifetime of a WSN with the proposed clustering method, the three metrics presented above are used: FND, HND, and LND.

As was mentioned previously, the rules of the KB can differently affect these three metrics (FND, HND, and LND). For example, one rule can increase FND, while it decrements HND or LND. Moreover, keeping the same weight for all the rules for the entire lifetime of the network may not be beneficial for all the metrics. Consequently, we propose a weight tuning for each rule for the three stages that the instants FND, HND, and LND delimit. Thus, we have three KBs, which are as follows:from the first round to FND, we use an initial KB.From FND to HND, we rely on an intermediate KB.From HND to LND, we work with the final KB.

Thus, to obtain the three different KBs, the weights of the rules are optimized following the process depicted in Figure 4.

According to Figure 4, to carry out the optimization process, the initial stage starts with all the rules with a weight of one, to obtain the best weights that maximize the value of FND, so we obtain the initial KB or $KB_{initial}$. It should be noted that, for the whole optimization process, we use a set of scenarios that define different WSN deployments with common parameters (the size of the deployment area, number of nodes, etc.) The details can be found in Section 2.4. In the second phase of the optimization process, $KB_{initial}$ is used as the initial KB, until the simulation reaches FND. After that point, the weights of the rules are optimized again to obtain the best HND. The resulting KB is called $KB_{intermediate}$. The third step of the optimization process again takes $KB_{initial}$ as the first until FND, then changes to $KB_{intermediate}$ from FND to HND, and then the weights are optimized to find the best LND to finally obtain a new KB named $KB_{final}$.

Therefore, the final clustering method is based on the three different KBs that are employed in different stages over the network’s lifetime. These KBs mainly differ in the weights of their rules. Moreover, due to the fact that the clustering method is centralized, a change in the KB in the fuzzy controller in the BS does not entail any energetic effort or complex operation. Once the three sets of weights have been obtained, they can be studied to interpret some of the proposed rules, taking into account the lifetime of the WSN and their role in the KB.

### 2.3. Energy Model

The optimization of the weights in the three KBs requires performing WSN simulations. In this process, it is required to model the energy spent in the communication of the nodes. In our case, this was based on the first-order radio model detailed in [41]. This well-known model defines a set of equations that take into account the energy spent in the data exchange between radio devices, like those installed in the nodes. In addition, it is a reference widely used in the related literature. The main equations of the model are shown in Equations (Equation 2)–(Equation 4). This energy model calculates the energy spent in the transmitter and in the receiver based on the length in bits of the data message and the distance between them, assuming free space and multi-path fading channels. Thus, the energy consumed by the transmitter is detailed in Equation (Equation 2), while the energy spent by the receiver is shown in Equation (Equation 4).
(2)$$E_{Tx}\left(l,d\right)=f\left(x\right)=\left\{\begin{array}{cc} l\cdot (E_{elec}+E_{fs}\cdot d^{2}), & d\leq d_{0}\\ l\cdot (E_{elec}+E_{mp}\cdot d^{4}), & d>d_{0} \end{array}\right.$$
(3)$$d_{0}=\sqrt{\frac{E_{fs}}{E_{mp}}}$$
(4)$$E_{Rx}\left(l\right)=E_{elec}\cdot l$$
where

*l* defines the number of bits of the message.$E_{elec}$ measures the energy in Joules that the circuitry of the transmitter and the receiver consumes for each bit sent or received, respectively.*d* is the distance in meters between the source and the destination of the message.$E_{fs}$ stands for the energy in Joules that the amplifier consumes in order to obtain a acceptable bit error rate, according to the free space model $\left(d\leq d_{0}\right)$.$E_{mp}$ is the energy in Joules that the amplifier consumes to obtain an suitable bit error rate in the multi-path (mp) model $\left(d\leq d_{0}\right)$.

This model also takes into account the reception of data and the aggregation process in every CH [42]. In a WSN aggregation, it is necessary to save energy by avoiding sending the value of each quantity measured by a node independently. Thus, it is a common practice that, through a statistical operation, only a summary of these values is sent from the CH to the BS, so that the energy spent in each bit of a received and aggregated message is defined as in Equation (Equation 5):(5)$$E_{Rx-DA}=\left(E_{elec}+E_{DA}\right)\cdot l$$
where

$E_{DA}$ stands for the energy in Joules spent by the CH when it receives and aggregates the data from a contributing node.*l* is the number of bits of the final aggregated message.

### 2.4. Simulation Scenarios and Network Model

As mentioned above, the process used three typical WSN scenarios to optimize the weights of the rules, which were also used by other clustering methods for comparison. The dimensions of the node deployment area were the same for the three variants: a square area of 100 × 100 m^2^, where nodes were randomly placed. The number of nodes deployed was fixed for all scenarios (100 nodes). The main difference between the three scenarios was the location of the BS, which can be observed in Figure 5. The three scenarios can be identified as follows:Scenario 1: BS located at (100, 0) m, a corner of the deployment area.Scenario 2: BS located at (150, 50) m, outside of the sensing area.Scenario 3: BS located at (50, 50) m, in the center of the deployment area.

In addition, the network model used in this approach specifies that each node transmits and receives through a symmetric communication channel without interference. The BS, CH, and contributing nodes used time division media access (TDMA) to send messages in a single-hop transmission directly to the destination. All messages containing information from normal nodes to CHs and from CHs to BS used the minimum power required to reach the destination based on the information the BS knew about the networks. The power consumed by the BS was not considered, because it was assumed that the BS had an unlimited power source. In addition, all nodes had the same initial energy. Finally, the values of the parameters used in the optimization process are shown in Table 2. As can be seen, these parameters belong to the first-order radio model described in Section 2.3.

### 2.5. Proposed Optimization Algorithm

As was mentioned in Section 2.2, in this paper, we used a scheduling algorithm that fine-tuned the weight of rules within a fuzzy rule-based system [43], in order to obtain three different KBs that tried to optimize the performance and lifetime of the WSN:$KB_{initial}$ that maximizes the value of FND.$KB_{intermetiate}$ that maximizes the value of HND.$KB_{final}$ that maximizes the value of LND.

The tuning process employed a particle swarm optimization (PSO) strategy, referred to as PSO rule weight tuning (PSO-RWT). The reason for using PSO was its fast convergence, which makes it suitable for and widely used in other clustering methods in WSN [44]). In addition the authors have extensive experience in its use in other optimization problems. PSO-RWT was utilized to enhance the performance of FRBS by optimizing the rules involved. Consequently, each rule $R_{i}$ was multiplied by a weight factor $w_{i}$ (Figure 6). PSO has proven its effectiveness, taking advantage of the stochastic evolution of swarm intelligence and updating particles based on their internal velocities. A similar optimization can be found in [39]. In our proposal, the fitness function was changed to match the sensor network metrics (FND, HND, and LND).

In PSO, each individual, called a ’particle’, navigates within a multidimensional space that represents the search area. The system starts with a set of *M* particles randomly distributed throughout the n-dimensional search space $R^{N}$. Additionally, a real-valued function *f* is defined within this space, serving as the fitness function, $f:R^{N}\to R$. This fitness function is defined by the maximum value obtained for the three instants that delimit the interval of use of any of the three new KBs. Thus, for the first KB, the fitness function is the maximum FND; for the second, the maximum HND; and for the third, the maximum LND. Throughout the iterations, the position of each particle is updated with the goal of finding the optimal position where the best value for the fitness function or optimal state is achieved. As mentioned earlier, we chose the maximum value as the fitness function criterion. This positional adjustment was controlled by the components detailed in Equations (Equation 6) and (Equation 7), which drove the process.
(6)$$v_{i}^{\left(t+1\right)}=d_{0}v_{i}^{t}+d_{1}r_{1}\left(Pb_{i}^{t}-x_{i}^{t}\right)+d_{2}r_{2}\left(Gb^{t}-x_{i}^{t}\right)$$
(7)$$x_{i}^{\left(t+1\right)}=x_{i}^{t}+v_{i}^{\left(t+1\right)}$$

Here, $d_{1}$ and $d_{2}$ are constant factors, $Pb_{i}$ is the best position that particle *i* has achieved, and $Gb_{i}$ denotes the best position found by the neighbors of particle *i*. The terms $r_{1}$ and $r_{2}$ are random factors within the [0, 1] interval, and $d_{0}$ is the inertia weight. To ensure the algorithm’s convergence, velocity values *v* are restricted to the interval [$v_{min}$, $v_{max}$]. Another critical factor for convergence is the efficient setting of the inertia weight $d_{0}$. A higher $d_{0}$ value promotes global search, whereas a lower value encourages local search. In other words, $d_{0}$ can be adjusted to balance both global and local searches, reducing the number of operations needed to find the optimum value. Typically, *w* is set to decrease over *i* iterations, intensifying local searches after the entire space has been explored. It is usually initialized with a value close to one. The Table 3 presents the values used for these parameters, assuming a number of 100 for generations (maxgen).

The proposed algorithm treats a particle as a weight set. A weight set is defined as the parameters that multiply each rule in a fuzzy knowledge base. Thus, a particle is structured as illustrated in Equation (Equation 8) (*i* is the number of particles and *n* is the number of rules).
(8)$$P_{i}=\left(\begin{array}{c} w_{1}^{i}\\ ...\\ w_{n}^{i} \end{array}\right)$$

The particles are randomly generated, within a limited workspace. Their dimension is fixed and is linked to the number of fuzzy rules. The particles change their position according to the velocity in Equation (Equation 1), considering that the velocity of each particle ($V_{i}$) is determined by Equation (Equation 6).
(9)$$V_{i}=\left(\begin{array}{c} V_{1}^{i}\\ ...\\ V_{n}^{i} \end{array}\right)$$

Throughout the PSO process, each particle changes its position in the search domain (workspace). Since each particle is composed of a set of rule weights, modifying the particle’s value implies that the set of rule weights has changed. If any of the weights are outside the [0, 1] interval, the value is adjusted. The process continues until the stop condition is reached.

The following algorithm (Algorithm 1) summarizes the proposed scheduling and tuning process based on PSO.
**Algorithm 1:** PSO-RWT algorithm              1. Swarm: Num_particles,                             Num_iter, Init_rate ($r_{0}$), Inertial_weight ω, Factors $c_{1}$ and $c_{2}$.              2. Random setting of swarm position.              3. Random setting of velocity.              4. Velocity constraints.              5. Initialize Gbest ($P^{*}$)/Pbest ($P^{\#}$)          **Do****              Do**                  1. Update position. Equation (Equation 7).                  2. Constraints RB-Swarm position.                             $0\leq |P_{i}|\leq 1$                  3. Evaluate-Fitness. Evaluation system.                  Particles++              **While** (Num_particles)                  Update Gbest ($P^{*}$).              **Do**                      1. Update Pbest ($P^{\#}$).                      2. Update velocity. Equation (Equation 6).                      3. Velocity constraints.                  Particles++              **While** (Num_particles)           iter++          **While** (Num_iter)          **Return** solution: Gbest ($P^{*}$)

The process detailed in the PSO-RWT algorithm was repeated three times to obtain the three different KBs enumerated at the beginning of this section. First, the PSO-RWT algorithm was applied for 30 different layouts taken from each of the three scenarios described in Section 2.4 (10 from each scenario), to obtain the weights that maximized the FND, which was the stop condition for this phase, obtaining $KB_{initial}$. Then, the PSO-RWT was repeated, taking $KB_{initial}$ as the starting KB, and optimizing the rule weights from the FND until a maximum HND was reached. The new rule weights defined $KB_{intermediate}$. Finally, the PSO-RWT algorithm was executed to obtain the maximum value for LND, using $KB_{initial}$ until the FND was reached, $KB_{intermediate}$ from FND to HND, and from HND to LND, and PSO-RWT optimized the rule weights to obtain the new $KB_{final}$. Therefore, the optimization process produced three different KBs, which were then used by the clustering algorithm in the three stages of the WSN lifetime.

Finally, when the optimization process ended, three different KBs were obtained: each one to be used in a different stage of the WSN lifetime, as mentioned in Section 2.2. Consequently, although three KBs were obtained, this does not mean that the clustering method had three variants, because each KB was only applied in one stage during the lifetime of the WSN. In the next section, we present a comparison of the proposed CFC3PSO clustering method with other algorithms, to contrast their behavior and performance.

## 3. Results

In order to check whether the optimization process presented in Section 2.5 was adequate for improving the performance of the proposed clustering method for the three types of scenarios described in Section 2.4, this section will detail the results for our approach CFC3PSO in comparison with three other well-known clustering methods and a recent one: (i) Low-Energy Adaptive Clustering Hierarchy (LEACH) [12], (ii) Cluster Head Election mechanism using Fuzzy logic (CHEF) [20], (iii) Clustering Routing protocol for WSN based on type-2 fuzzy logic and ant colony optimization (CRT2FLACO) [45], and (iv) Intelligent Clustering Under Uncertainty (ICUU) [33], which is the newest method. Although LEACH is an old distributed method, it is still used as a pattern for comparisons because it represents a simple mechanism for clustering that only employs stochastic methods to form the cluster. Thus, in LEACH, each node can independently become a CH based on an initial probability and the number of rounds. In CHEF, the authors used a fuzzy system to select the CHs. This fuzzy system has two input variables: the residual energy of each node, and the addition of the distance to every node that is within a predefined range. The third algorithm, CRT2FLACO, uses a centralized Type-2 fuzzy controller with three input variables: the residual energy of each node, the distance between the node and the BS, and the number of neighbors of a node. In addition, it employs ant colony optimization (ACO) to decrease the transmission energy from the CH to the BS. The last method in the comparison was ICUU, which proposes an intelligent clustering mechanism using the silhouette index (SI) score, which is later used as a benchmark for optimized clustering conducted with the Density-Based Spatial Clustering of Applications with Noise (DBSCAN) algorithm.

The results were obtained by simulating the five methods (CFC3PSO, LEACH, CHEF, CRT2FLACO, and ICUU) for thirty WSNs generated for each scenario detailed in Section 2.4 that we call a map. A map contains the records of the locations of the 100 randomly deployed nodes, the position of the BS, and the value of the parameters of the energy model (Section 2.3. We then ran each method for the 30 different maps to obtain the FND, HND, and LND metrics, and finally obtained the mean of each of these metrics for the scenario under test. The simulations were performed in Matlab. The results for each scenario can be seen in Table 4 for scenario 1, where BS was located in a corner of the deployment area; Table 5, with the BS placed outside the area at coordinates (150, 50); and Table 6, with the BS in the center of the area.

As can be seen in Table 4, the proposed method CFC3PSO achieved better results than any of the compared methods for all metrics. Due to the optimization of FND, this metric for CFC3PSO was much better than the others, which suggests that the WSN could operate with all its nodes for longer than any of the other presented approaches. In addition, there was an increment in the value for HND, which was even higher than the LND of two other methods and very similar to the LND of CRT2FLACO and ICUU. Thus, these results show that the optimization performed clearly outperformed the other methods. Finally, the value obtained by CFC3PSO for LND was higher than any other approach. However, it was close to the HND, which means that the nodes had a very low energy after maximizing the HND and consequently they could not live much longer.

In Table 5, we can again see that CFC3PSO outperformed all the other methods compared, with a better value for FND, which was even higher than the LND values of the other approaches. In this case, the values obtained by CFC3PSO for HND and LND were slightly lower than those in scenario 1, due to the distance to the BS, which was located outside the deployment area of the nodes.

For this last scenario (Table 6), the improvement in the metrics caused by CFC3PSO was clearly significant. For FND, the value obtained was more than twice that of LEACH or CRT2FLACO, while it was 30% greater than with CHEF and significantly greater than with ICUU. In this case, the difference between HND and LND was greater than in the other scenarios. Considering HND and LND, CFC3PSO achieved the best results in the three cases because of the location of the BS in the center of the area.

In conclusion, we could establish that with the proposed clustering method CFC3PSO, a WSN that uses a fuzzy controller with an optimized KB for the three stages can achieve better metrics than the other approaches, even those with more complex methods like ICUU or CRT2FLACO, which use a Type-2 fuzzy system (our approach only needs a Type-1 fuzzy system, which requires less computational resources) and an ACO.

In the following subsection, we present the results of applying the previous clustering methods and the proposed CFC3PSO to two new WSN designs, to test the adaptability of the latter method to BS locations other than those used in the optimization process.

### Comparison with New Scenarios

The purpose of this section was to validate the adaptability of the proposed CFC3PSO clustering method to WSN deployments that were not included in the optimization process. Therefore, Table 7 and Table 8 show the results for a new set of simulations with the BS located at coordinates (75, 75) and (125, 50), respectively. All other parameters remained unchanged (see Section 2.4).

As can be seen from the results in both tables (Table 7 and Table 8), the proposed method CFC3PSO continued to perform better than any other tested algorithm, significantly improving on the two more complex ones (CRT2FLACO and ICUU). Therefore, the proposed approach proved to be a good solution even for WSN deployments not included in its optimization.

## 4. Rule Weights Interpretation

This section aims to compare the weights obtained from the optimization process, to analyze them and give a proper interpretation of the three metrics (FND, HND, and LND) and of the three scenarios used. The first part of this section will show some examples of the differences among the weights comparing FND with HND, FND with LND, and finally HND with LND. The last part will present some of the rules that changed their weight depending on the location of the BS to obtain better metric results.

### 4.1. Metric Comparison

This section presents some examples of rules that received weight values that can be clearly contrasted because they were 0 or 1. Therefore, these values assume that a rule was not evaluated at all (weight 0) or that it was completely meaningful for this stage (weight 1). Note that the weight values for all other rules were between 0 and 1 and can be found at the published dataset (See data availability statement).

In Table 9, three rules can be found that show a significant difference between the weights for FND and HND when the BS was located in the center of the deployment area (Scenario 3). The table shows the weights obtained for $KB_{initial}$ or $W_{initial}$ and the weights for $KB_{intermediate}$ or $W_{intermediate}$.

In this case, the rules in Table 9 were not taken into account in the KB to obtain the best FND, because their weights were 0. However, they should not be removed from the KB because they are needed to maximize the HND in the second stage. In a similar way, in Table 10, the weights for the other three rules can be found (take $W_{final}$ as the weight obtained from the optimization of $KB_{final}$), which had differences in the optimization process. In this case, the three rules were meaningful to maximize LND, whereas they were not taken into account to obtain the best FND.

The third example of differences in the weights optimized by the PSO-RWT algorithm are shown in Table 11. In this case, the four rules in the table were used to maximize the HND, while they were not considered for the LND.

One conclusion of this first analysis of the results is that fuzzy systems are difficult to design, even more when they have to use a high number of variables with complex layouts. In this case, rule optimization can help to tune the knowledge base and optimize the results. Consequently, as can be seen in Table 12, the knowledge base may sometimes contain rules that are useless for some situations, such as the rules when the base station was located in a corner of the coverage area (scenario 1). However, these rules cannot be removed from the KB, as previously mentioned, because they may be useful for other situations.

### 4.2. Scenario Comparison

After analyzing the weights obtained for maximizing FND, HND, or LND, in this section, we present some examples of rules that could improve the result for one metric in some scenarios, but not for all, while there were other rules that maximized one metric for all the scenarios.

In Table 13, all the rules that improved the results for FND only for Scenario 1 are enumerated, whereas they were completely useless for the other two.

Moreover, we can find examples of rules that maximized the result of FND for two scenarios (see Table 14) or for all, as can be seen in Table 15. In this last case, it can be observed that, for example, for most rules, the node was at an “adequate” distance from the BS, which means that nodes placed at an intermediate distance help the performance of the network to achieve good metrics. In contrast, in Table 16, the detailed rules were completely useless to maximize FND. We could make similar observations about the other two metrics, HND and LND, but we have summarized the results for FND because the whole set can be found in the link at the beginning of Section 4.1. We should remark that, after the optimization process, there were no rules whose weight was 0 for the three scenarios, consequently, the application of the three stages of KB had no rules without importance for this clustering method.

## 5. Conclusions

The presented CFC3PSO clustering algorithm was shown to be able to handle three different WSN layouts and achieved better results than the four other clustering algorithms. Consequently, this work proved the premise that a three-stage fuzzy controller based on three different knowledge bases optimized by a PSO can improve on the performance of other clustering methods, without drastically increasing the complexity. Furthermore, the rule weights obtained by the optimization process effectively added a new layer of information to the initial rule base, allowing a more precise definition of the rules for future work. More specifically, first, the presented optimization process based on the PSO-RWT algorithm successfully obtained three knowledge bases by optimizing the weight of its rules to maximize the FND, HND, and LND metrics for three different scenarios, in order to achieve better adaptability for new WSN designs that could have their BSs in different locations than those used in the optimization process. In addition, another contribution of our work was the way in which the three KBs were obtained by defining a three-stage process, where the first KB or $KB_{initial}$ is applied by the FRBS until the first node dies, then the second KB or $KB_{intermediate}$ is loaded in the fuzzy engine until HND is reached, and finally using the third KB or $KB_{final}$ to obtain the maximum WSN lifetime.

Overall, in this paper, we presented a centralized clustering method that minimizes the communication between nodes based on a type-1 FRBS whose optimized KBs vary during the three stages. This proposal achieved significantly better results than the other compared methods and suggested an improvement due to the tuning of the knowledge base by the PSO-RWT algorithm, which detected which rules were more or less meaningful for each stage and scenario.

As future lines of work, we propose changing the FRBS from a Mamdani to a Takagi–Sugeno–Kang type [46], whose functional output is suitable for optimization by the PSO-RWT algorithm. Moreover, we could explore additional design parameters for the presented approach, such as increasing the skip from 1 to a higher value, which could reduce the number of transmissions from the BS to the nodes in order to reduce the reception energy cost, taking into account that a too high skip value reduces the CH rotation, which could lead to premature deaths, so this parameter is also suitable for being optimized. In addition, the results obtained after the optimization process, in terms of the weights for each rule of the KB, could be tested under other WSN conditions, to check how these weights improved the lifetime of other WSN layouts.

## Figures and Tables

**Figure 1 sensors-24-05548-f001:**
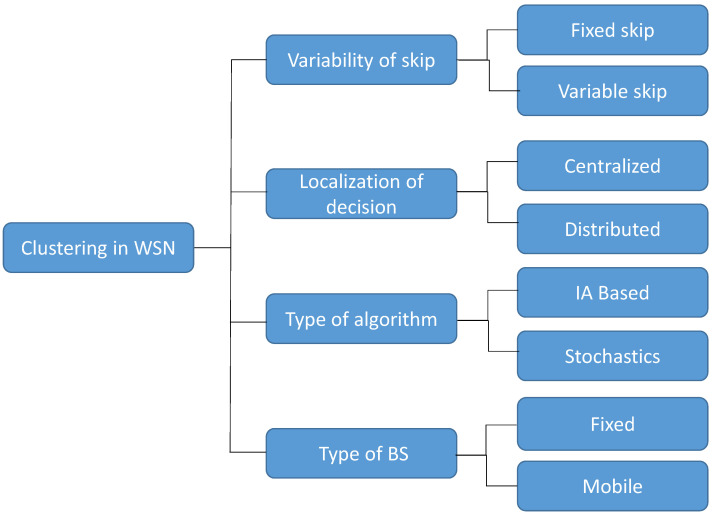
Clustering algorithm classification in WSN applications.

**Figure 2 sensors-24-05548-f002:**
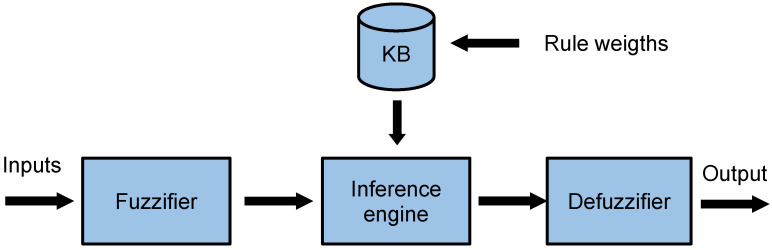
Fuzzy inference system.

**Figure 3 sensors-24-05548-f003:**
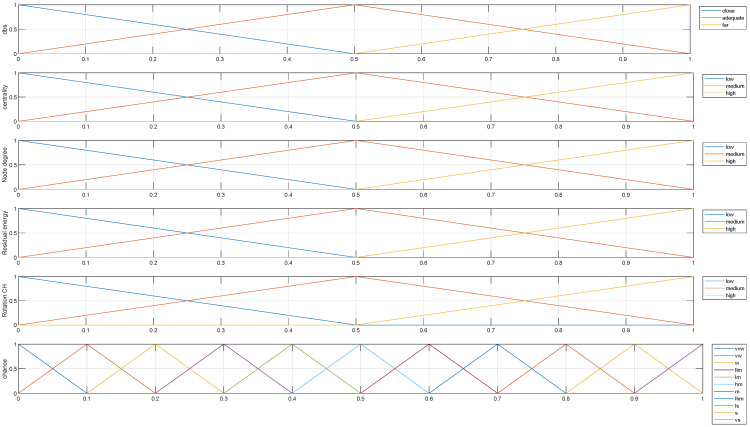
Fuzzy set layout for our clustering method.

**Figure 4 sensors-24-05548-f004:**
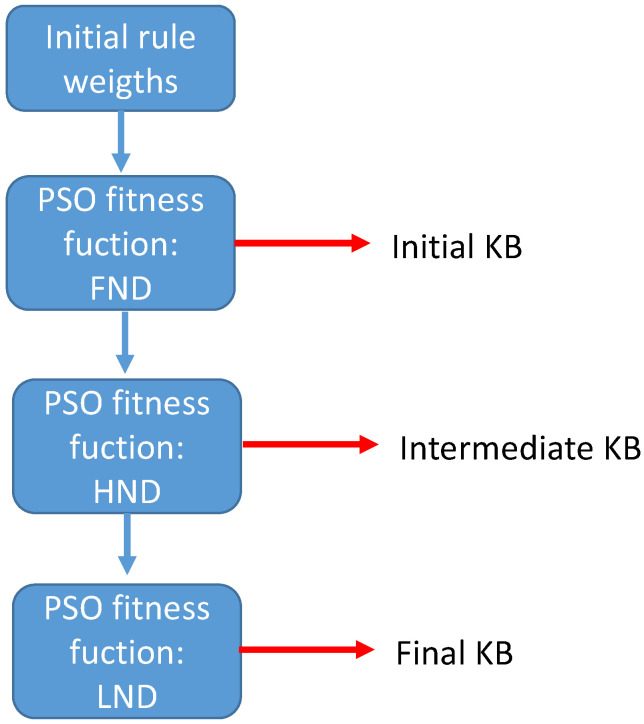
Optimizing the rule weights for the three KBs.

**Figure 5 sensors-24-05548-f005:**
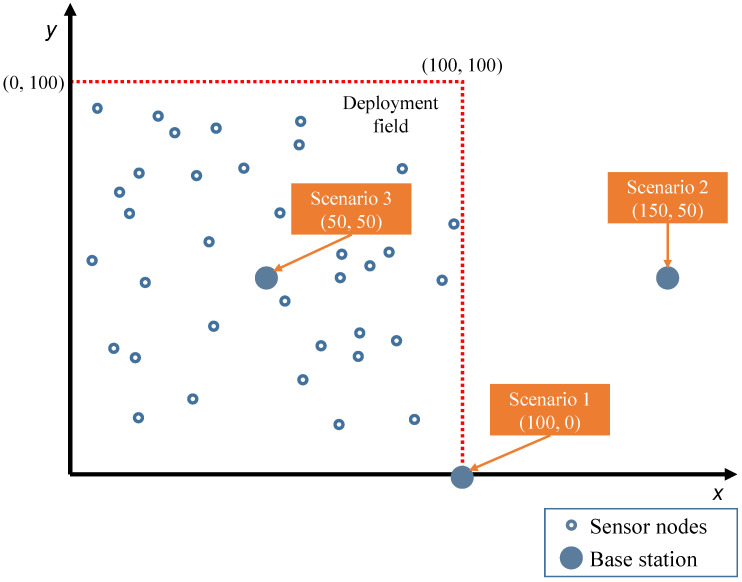
Simulation scenarios.

**Figure 6 sensors-24-05548-f006:**
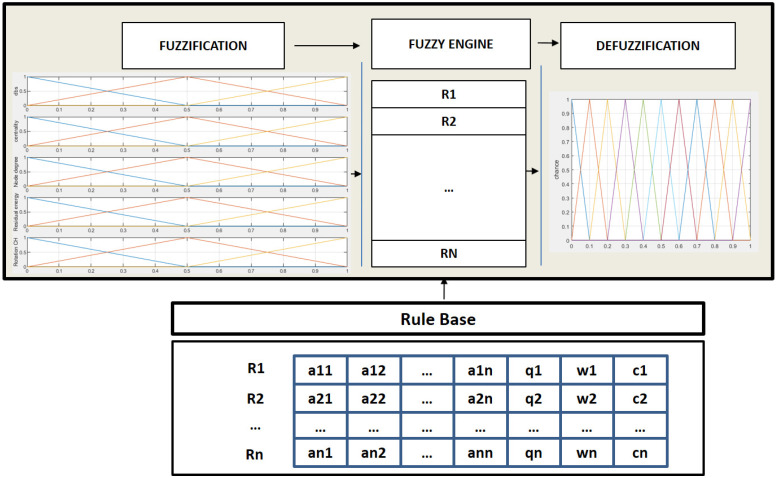
Fuzzy rule-based system.

**Table 1 sensors-24-05548-t001:** Samples of rules in the KB.

dBS	C	ND	RE	CHR	Weight	Output
C	L	L	L	M	1	llm
A	H	H	M	M	1	ls
F	H	H	H	H	1	s

**Table 2 sensors-24-05548-t002:** Values of the setup parameters for the energy model.

Parameter	Value
Initial energy of nodes	0.5 J
Length of control message	200 bits
Length of data message	2000 bits
$E_{elec}$	50 nJ/bit
$E_{DA}$	5 nJ/bit
$E_{fs}$	10 pJ/bit/m^2^
$E_{mp}$	0.0013 pJ/bit/m^4^

**Table 3 sensors-24-05548-t003:** PSO parameters.

$d_{0}$	$d_{1}$	$d_{2}$	maxgen
$e^{-\frac{1}{{\left(i*maxgen\right)}^{2}}}$	1	2	100

**Table 4 sensors-24-05548-t004:** Results with the BS located in the corner of the deployment area.

Algorithms	FND	HND	LND
LEACH	754.2	905.3	1002.1
CHEF	1414.3	1766.3	1827
CRT2FLACO	671.3	1596.8	2148.4
ICUU	1397	1900.7	2115.9
CFC3PSO	1827.3	2117.7	2222.5

**Table 5 sensors-24-05548-t005:** Results with the BS located at coordinates (150, 50) m.

Algorithm	FND	HND	LND
LEACH	713.1	841.03	994.46
CHEF	1379.4	1753.1	1785.5
CRT2FLACO	604.1	1479.4	1837.4
ICUU	1231.9	1769.6	2040.6
CFC3PSO	1868.5	1972.7	2067.4

**Table 6 sensors-24-05548-t006:** Results with the BS placed in the center of the deployment area.

Algorithm	FND	HND	LND
Algorithm	FND	HND	LND
LEACH	764.8	936.5	1020
CHEF	1402.2	1767.9	1827.2
CRT2FLACO	510.9	1713.4	2183.4
ICUU	1502.7	1984.7	2153.1
CFC3PSO	1847.2	2198.1	2344.7

**Table 7 sensors-24-05548-t007:** Results for the BS placed at x = 75 m and y = 75 m.

Algorithm	FND	HND	LND
LEACH	764.1	931.2	1019.4
CHEF	1403.6	1775.7	1816.7
CRT2FLACO	749.5	1707.6	2199.1
ICUU	1485.3	1966.4	2145
CFC3PSO	1800.6	2116.8	2359.6

**Table 8 sensors-24-05548-t008:** Results with the BS placed at x = 125 m and y = 50 m.

Algorithm	FND	HND	LND
leach	744	897.1	1007.5
chef20	1403	1769.3	1811.4
CRT2FLACO	639	1602.2	2040
ICUU	1366.3	1885.7	2097.4
CFC3PSO	1808.3	2064.1	2223.9

**Table 9 sensors-24-05548-t009:** FND versus HND. BS in center.

dBS	C	ND	RE	CHR	Output	$W_{\mathrm{initial}}$	$W_{\mathrm{intermediate}}$
C	H	L	H	L	hm	0	1
A	L	H	M	M	m	0	1
C	H	L	H	M	ls	0	1

**Table 10 sensors-24-05548-t010:** FND versus LND. BS in center.

dBS	C	ND	RE	CHR	Output	$W_{\mathrm{initial}}$	$W_{\mathrm{final}}$
C	M	H	M	L	hm	0	1
F	H	L	M	H	m	0	1
F	M	M	M	H	m	0	1

**Table 11 sensors-24-05548-t011:** HND versus LND. BS in center.

dBS	C	ND	RE	CHR	Output	$W_{\mathrm{intermediate}}$	$W_{\mathrm{final}}$
F	H	M	M	H	hm	1	0
A	L	H	H	H	ls	1	0
C	M	M	L	M	m	1	0
C	L	M	L	L	llm	1	0

**Table 12 sensors-24-05548-t012:** Rules whose weights were null for the three KBs for Scenario 1.

dBS	C	ND	RE	CHR	Output
F	M	H	L	L	llm
C	L	M	M	L	lm
C	L	L	H	L	lm
A	L	H	L	M	lm
F	M	M	M	M	lm
A	L	H	M	M	m
C	H	M	H	M	s
F	L	L	M	H	llm
A	M	L	H	H	hm
C	M	M	H	H	s

**Table 13 sensors-24-05548-t013:** Rules that optimized FND for Scenario 1 but not for 2 and 3.

dBS	C	ND	RE	CHR	Output
F	M	M	H	L	lm
A	M	L	L	M	llm
A	L	H	L	H	m
C	M	H	M	H	s
A	M	H	M	H	ls

**Table 14 sensors-24-05548-t014:** Rules that optimized FND for Scenario 1 and 2 but not for 3.

dBS	C	ND	RE	CHR	Output
dBS	C	ND	RE	CHR	Output
F	M	M	L	L	w
A	L	L	H	L	llm
C	L	H	M	M	hm

**Table 15 sensors-24-05548-t015:** Rules that optimized FND for all the scenarios.

dBS	C	ND	RE	CHR	Output
A	L	H	L	L	llm
A	H	L	H	L	m
A	H	M	H	L	hm
F	M	L	L	M	w
A	H	M	H	M	ls
A	L	H	H	M	hm

**Table 16 sensors-24-05548-t016:** Rules that were not meaningful for FND for all the scenarios.

dBS	C	ND	RE	CHR	Output
A	L	H	L	L	llm
A	H	L	H	L	m
A	H	M	H	L	hm
F	M	L	L	M	w
A	H	M	H	M	ls
A	L	H	H	M	hm

## Data Availability

The dataset can be found in https://ruja.ujaen.es/jspui/handle/10953/3168 (accessed on 28 July 2024) of the public Institutional Repository of Scientific Production of the University of Jaén.

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
