# Peer review of "Optimizing Rule Weights to Improve FRBS Clustering in Wireless Sensor Networks"

_sensors, 2024, doi:10.3390/s24175548_

Round 1

Reviewer 1 Report

Comments and Suggestions for Authors

In this article, the weight of each rule of the fuzzy logic controller used to choose the CH is optimized through a Particle Swarm Optimization for the three instants. Some issues should be addressed as follows.

1.   In the abstract, the introduction of the background should be simplified, and the novelty, motivation, and main contributions should be better summarized.

2.   As the background of this paper includes wireless sensor networks (WSNs), Internet of Things (IoT), and data transmission between nodes and base stations, recent high quality works on the relevant research areas should be introduced in the first paragraph of the introduction, such as Distributed DDPG-based resource allocation for age of information minimization in mobile wireless-powered Internet of Things, IEEE IoTJ, 2024. More references about the background of this paper should be introduced.

3.  The first paragraph in the introduction is suggested to be divided into more paragraphs to improve the logic of this paper.

4.   Authors need to explain the statement that “Thus, regarding skip, most of the revised papers set a constant value of one”.

5.   The motivation in the introduction is not clear, and the main contributions should be better summarized in terms of technical contributions.

6.  Authors need to specify why 11 triangular fuzzy sets is considered in this paper. How the number of the triangular fuzzy sets is determined?

7. Why authors adopt Particle Swarm Optimization for optimization? How does the proposed clustering algorithm advance in the research field?

8. Conclusions should be better summarized in terms of the main contributions and results.

Comments on the Quality of English Language

Minor editing of English language is required for this manuscript.

Reviewer 2 Report

Comments and Suggestions for Authors

To achieve WSN goals,this paper propose a algorithm combined with Particle Swarm Optimization to help the system select CH.The results of the presented approach improve significantly upon those from other authors with similar methods

As to this paper, I have some comments as follows.

1.In Abstract, I suggest adding an explanation of the abbreviation before "WSNs" appears 

2.In Abstract, I suggest a more detailed description of the methods proposed in this paper. 

3.in Section 2.5, the fitness evaluation function f should be explained in more detail, because it is very important for the PSO algorithm. 

4.In section 4, is it possible to add more new algorithms for comparison? Since the latest compared algorithm was published in 2015.

Comments on the Quality of English Language

To achieve WSN goals,this paper propose a algorithm combined with Particle Swarm Optimization to help the system select CH.The results of the presented approach improve significantly upon those from other authors with similar methods

As to this paper, I have some comments as follows.

1.In Abstract, I suggest adding an explanation of the abbreviation before "WSNs" appears 

2.In Abstract, I suggest a more detailed description of the methods proposed in this paper. 

3.in Section 2.5, the fitness evaluation function f should be explained in more detail, because it is very important for the PSO algorithm. 

4.In section 4, is it possible to add more new algorithms for comparison? Since the latest compared algorithm was published in 2015.

Round 2

Reviewer 1 Report

Comments and Suggestions for Authors

Authors have well addressed my previous comments.

Reviewer 2 Report

Comments and Suggestions for Authors

My previous comments have been addressed.